# An Agent-Based Simulation to Minimize Losses during a Terrorist Attack

**Ondrej Dolezal [†] and Hana Tomaskova [*],[†]** 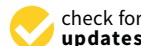

Faculty of Informatics and Management, University of Hradec Kralove, Rokitanskeho 62,
50003 Hradec Kralove, Czech Republic; ondrej.dolezal@uhk.cz
* Correspondence: hana.tomaskova@uhk.cz
† These authors contributed equally to this work.

**Abstract:** The goal of this paper is to create a model simulating a part of the terrorist attack in the Tokyo subway in 1995 using sarin gas and its implementation using AnyLogic software. Another goal is to find possibilities of minimizing the losses using what-if scenarios. The model should serve as a tool for further investigation of the attack and proposes the appropriate security options in the future. The final part of the work summarizes the results of the research and suggestions for improving the model.

**Keywords:** agent-based model; AnyLogic; terrorist attack; simulation

## 1. Introduction

In 1995, an act unprecedented in Japan was committed in the capital of Japan (Tokyo). Five members of the terrorist group Aum Shinri Kyo released sarin gas during the rush hour in a total of five train lines. The consequences were devastating. More than 5000 people were injured, and 12 died from sarin exposure. Late evacuations and closures had a significant impact on the high losses. This took place only after several minutes. Symptoms of sarin infection, however, occur much earlier [1–4]. This act has been the subject of many publications [5–9], while publications [10–12] focused on evacuation and sarin gas was dealt with in publications [13–15].

Sarin is a highly toxic substance that is colorless, tasteless and odorless in its pure form. The US Center for Disease Control and Prevention (CDC) describes it as a human-made chemical weapon. The International Convention has banned the manufacture of a substance since 1993. In its basic form, it is a liquid that evaporates as gas and spreads through the air. In its purest form, it is estimated to be up to 26 times more deadly than cyanide [16].

Exposure to sarin is often fatal even at small concentrations. Sarin acts quickly through the respiratory and ocular system. The risk of infection depends on the duration of exposure and the dose to which the person is exposed. Since the substance is odorless and invisible, it is difficult for humans to detect its presence in the air. In addition to the airways, sarin is also absorbed through the skin, but this is much slower. The effects of exposure are usually in the order of a few seconds to minutes. Common symptoms of infection include sudden runny nose, eye pain, excessive salivation, difficulty breathing, sudden sweating, nausea or cough. More substantial exposure may lead to loss of consciousness, lung failure and death.

The lethal dose of sarin is most often reported as 100 mg-min/$m^3$. This means that 50% of the population will die from a one-minute stay in a place where the sarin density is 100 mg per cubic meter. Thus, if the density at the site of residence is, for example, 10 mg/$m^3$, half of the exposed population will die in ten minutes of residence at that location [17].

The rest of the paper is organized in the following manner: In Section 2, Related Works, we talk about the basic models that inspired the underlying logic of our model. In Section 3, we described the assumptions of the model. Section 4 describes the model in detail, according to [18]. The section begins by explaining the purpose of the model and continues with a description of the variables, processes and concept of the model. The following is a description of the initialization and input values. Section 5 talks about experiments performed on the model. These are three different scenarios, the results of which are described and were compared using statistical tests. We also presented the validation of the model with available information, as recommended in publications [19–21]. Section 6 discusses the possible limitations and extensions of the presented model, and in the last part, Section 7, we summarize the primary results and benefits of the text.

## 2. Related Works

The problem of attacks is relatively numerous for agent-based modeling environments. Therefore, it was possible to find basic models that could be incorporated into the final model, or at least be inspiring for the created model.

### 2.1. Terrorist Attack on a City Square

A sample model that is part of the AnyLogic installation package itself. The model is to simulate a fictional terrorist attack on a square with a large concentration of pedestrians. The pedestrian library was used for this purpose. It is one of the extension libraries that makes it much easier to simulate pedestrians and their behavior.

The model shown in Figure 1 is straightforward and contains only two types of agents—a pedestrian and a terrorist, with pedestrians as the population and a terrorist as only one agent. Pedestrians move freely around the square. After a while, a terrorist rides a truck in the square crosses several pedestrians along the way, and as soon as he arrives at the center of the square, where there is the highest concentration of pedestrians, he unlocks a bomb, killing everyone in the circuit. The survivors panic and set off as soon as possible to the nearest exit from the square. The model could, of course, be improved and expanded in many ways, but it is sufficient for demonstration purposes.

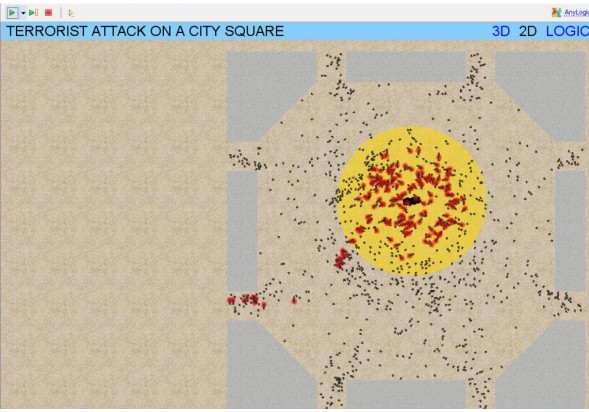

**Figure 1.** Sample model "Terrorist Attack on a City Square".

### 2.2. Emergency Evacuation in a Railway Station Simulation

Linag Ma et al. [10] present an agent-based oriented model of the evacuation of a railway station square during the same terrorist attack as in our study. The authors put a lot of effort into accurate agent behavior, which is based on the BDI (Belief–Desire–Intention) model, human behavior during stressful situations like emergency evacuation, and crowd dynamics in limited space.

*2.3. Wildfire*

The Wildfire model, as shown in Figure 2, is already a more advanced model in which the propagation of fire in the forest is modeled. An exciting feature of this model is that both continuous and discrete environments are used. This approach has been used in the practical application presented in this paper. A discrete environment is a grid, where each cell has a defined X and Y position. This environment is used to propagate fire in a sample model. Each box is an agent. As soon as a fire occurs, the lighted box always sends a message to all the tiles in the neighborhood, and the box also responds to the fire. The model has also included the influence of the wind on the fire.

The continuous environment here represents the sky in which a single agent is a moving airplane. This setting also affects the discrete environment. The aircraft is moving only in its continuous environment, but at some point, it fires a bomb and thus begins the initial spread of fire. The address of the field hit by the bomb must, therefore, be monitored. This field then starts spreading and sends a message to its surrounding areas. The model is well described in The Big Book of Simulation Modeling [22], including the source code.

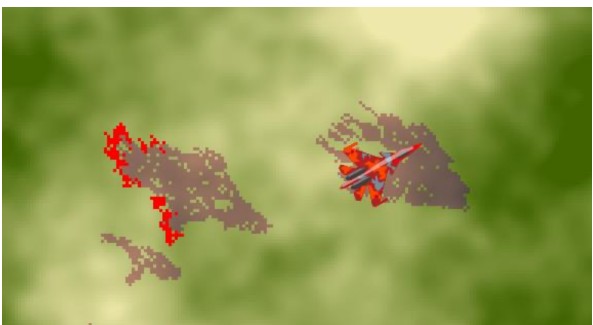

**Figure 2.** Sample model "Wildfire".

## 3. Model Assumptions

*3.1. The Amount of Sarin in the Attack*

In the attack in Tokyo used packages of sarin, one of which contained about 0.45 L of substance. Four attackers were equipped with two containers and one with three. Each attacker threw the parcels to the ground and pierced them with sharpened umbrellas before getting off the train. Fortunately, the attackers failed to puncture all the packages. Moreover, they did not use completely clean sarin and thus had about 30% of the average sarin concentration. This also made the gas smell. If the sarin was in pure form, the consequences would have been many times worse.

*3.2. The Course of the Attack*

On 20 March 1995, five members of the cult of Om Shinrikjo released sarin gas in the Tokyo subway. It is one of the busiest metros in the world. The attack occurred shortly after 8 A.M. when there is a morning rush in Tokyo. The attackers carried packages of sarin and a sharp-edged umbrella to puncture them. They got on pre-selected trains and dropped bags at the selected stations, pierced them and got out of the train. At the exit of the subway, accomplices in a car were already waiting for them to help them escape from the crime scene while the sarin was flying through several subway lines.

*3.3. Kodemmacho Station*

The Hibiyo Kodemmacho Subway Station, Figure 3, proved to be the place with the most significant losses. Yasuo Hayashi, who was the only one to carry three sarin packs, got on the train and managed to puncture two of the three packages. After several passengers on the train got sick, one of the passengers noticed, at Kodemmacho Station, containers on the ground. Packages from

the train were kicked directly to the platform, which later had fatal consequences. There were four deaths and hundreds of people exposed to sarin.

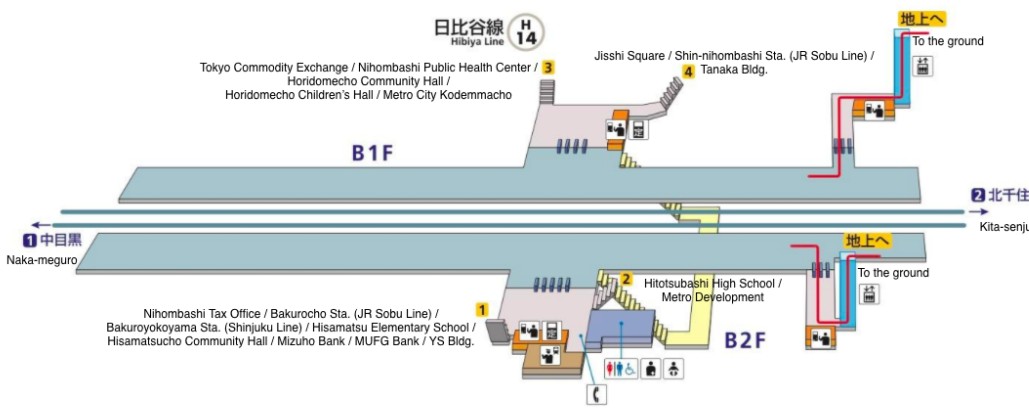

**Figure 3.** Map of the station Kodemmacho (with English labels).

## 4. Model

According to publication [18], the following part is divided into sections corresponding to the standard protocol for describing individual-based and agent-based models.

### 4.1. Purpose

The model aims to model as accurately as possible part of the terrorist attack in the Tokyo subway. Kodemmacho station, where the sarin packages were kicked out onto the platform, is modeled. The model should be as close to a real attack as possible. Subsequently, what-if scenarios will be developed to minimize losses in the event of a similar attack in the future and improve experiments based on these scenarios.

### 4.2. State Variables and Scales

The fundamental entity in the model is the person who represents the metro passenger. The model will thus capture its behavior from entering the station to boarding the train, or vice versa from getting out of the train to getting out of the station. The "Disease" variable is the human exposure value. It is calculated by the duration of exposure and the density of the sarin at its current position. The variable "Speed" determines the speed of human walking. It reaches values of $1 - 2m/s$, according to the average rate one achieves. Another significant entity is "Air". Air represents individual cells of a discrete environment, each cell being an agent. The "Distance" variable is a value that determines the Euclidean distance of a given cell from the cell in which propagation has occurred. The "Sarin" entity determines the actual density of the sarin in the cell. The "Concentration" variable determines the maximum value of sarin that a cell can reach given a distance from the source. The value is calculated based on the function obtained from the report from French intelligence [23].

### 4.3. Process Overview and Scheduling

Model time is a combination of asynchronous and synchronous, which means that events happen at any moment. Still, some events occur discreetly; for example, gas propagation takes place at specific time steps because the air environment is also discrete. Trains and people move continuously based on a given speed of movement, with the time unit of the model being a second concerning the required accuracy. Disease calculation of each person also takes place in discrete steps. One hundred times per second, the human position is checked, and the infection is calculated based on the known lethal sarin value based on the sarin density at the site.

### 4.4. Design Concepts

The model simulates a real metro station in Tokyo (Kodemmacho). The station is modeled on real scale based on data provided by the metro dispatchers themselves via email communication. The trains run on a real-time schedule, with the model starting at 8:00 A.M. when the attack was committed. The duration is set at 35 min, after which time the entire Hibiyo line was closed and people evacuated.

The first train arriving at the station will cause the source of the sarin to be kicked off the platform and the start of gas to air. The spread of sarin occurs in a discrete environment. One cell is a source of infection, and the gas continues to spread to cells in its eight connected neighborhoods. It is modeled using a state diagram where each cell can acquire an infected/uninfected state. If a cell gets infected by a received message from another cell, internal transit in that state causes it to spread to other cells by sending messages to its neighbors. In the infected cell, the density of the sarin is continuously added based on its distance from the source until the density value is equal to the maximum concentration calculated by function.

People appear to be classic passengers who either arrive at the metro station and wait for the train, or arrive at the station by train and leave the station by the nearest exit. Their behavior is mainly based on empirical data. The entrance to the metro is subject to the passage of the turnstile. At the same time, part of the population will be delayed by scanning the ticket (max. 3 s) and some passing immediately (assuming a prepaid card). The number of people at the station is influenced by the fact that the attack occurred in the morning rush hour and calculated from real values according to the average increase in the morning rush hour [24]. The station is set to the default number of people, as the model starts under regular metro operation. Since only the infection of individual people is monitored, people do not communicate with each other in any way.

One hundred times per second, the position of each person is checked, and the degree of infection is added based on the density of the sarin in the area where it is. The calculation formula is determined from the known lethal dose of sarin, which is 100 mg-min/m$^3$. Thus, if there is a 100 mg/m$^3$ density in the box where a person is present, the person will reach the lethal value in one minute. If the value is smaller, the amount is recalculated. The time step is designed so that the model is as accurate as possible, but at the same time, it takes into account computational demands.

People are then divided into three groups based on the degree of exposure.

- Light exposure—miosis, mucus in the nasal cavity, bronchoconstriction, choking. Marked in white in the model.
- Severe exposure—miosis, mucus in the nasal cavity, bronchoconstriction, heavy choking, loss of consciousness, convulsions, seizures, spontaneous muscle twitching, weakness etc. Marked in yellow by model.
- Lethal exposure—with a so-called lethal dose of sarin, 50% of the population exposed to this dose will die. In experimentation, the total value of the infected is divided by two. Marked in red in the model.

Given the purpose of the model, only the inhalation is taken into account. Absorbed through the skin, the risk is many times smaller and the infection would last considerably longer than the duration of the model, so it is not included in the model.

### 4.5. Initialization

In the initial state of the model, i.e., at time $t = 0$, there are several people stationed by default waiting for the train to arrive, as seen in Figure 4. Time $t = 0$ corresponds to 8:00; it means modeled attack time. The number of people is given randomly in the interval of 50–100 inclusive. No cell is yet a source of sarin distribution. Every cell representing air has a sarin value of zero, and human infection is therefore zero. The value of the cell spacing is zero due to the presence of the propagation source.

Initialization is always the same. Initial values of internal variables are selected based on accurate data. The default number of people is estimated based on the known hourly number of people.

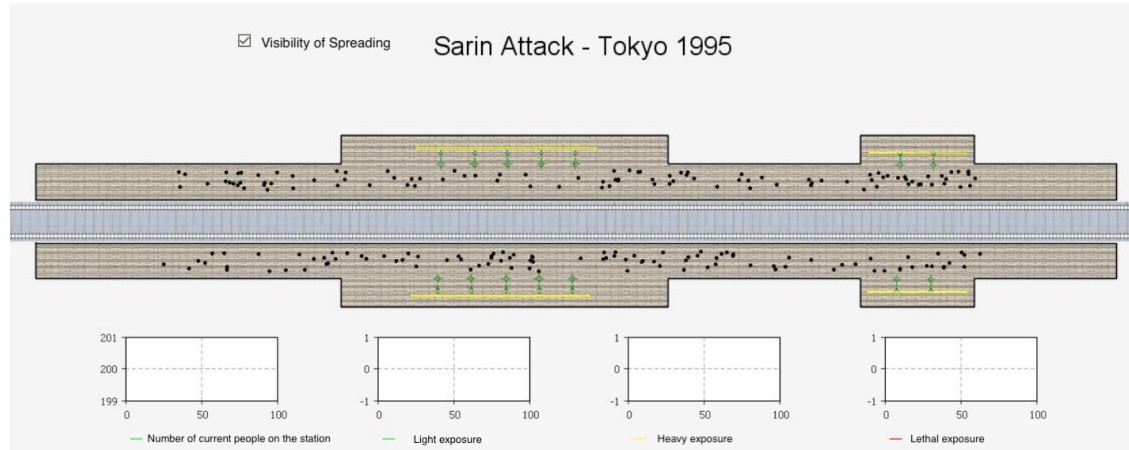

**Figure 4.** Model state at time t = 0.

As input data of the model, it is necessary to include the ratio of people at the station during the morning peak to the number of people off-peak. For more accurate results, a realistic timetable for departure and arrival of trains at the station is also included. Furthermore, it is necessary to know information about the spread of sarin, such as the density of the substance based on the distance from the source of range. Also important is the information on the lethal value of the sarin from which the disease is calculated.

The default model reproduces the real attack as accurately as possible. Since information about the total number of injured at Kodemmacho is not traceable, the accuracy of the model was judged by the number of deaths known. The averaging of one hundred simulations of the initial model yielded the results shown in Table 1, where the individual numbers correspond.

**Table 1.** Average model defaults.

|  | Light Exposure | Severe Exposure | Lethal Exposure | Original Fatalities |
|---|---|---|---|---|
| Average values | 499.43 | 11.22 | 5.55 | 4 |
| Maximal values | 559 | 19 | 11 | 4 |
| Minimal values | 446 | 5 | 2 | 4 |

*4.6. Input*

The model requires accurate data regarding number of passengers coming in and out of the station. Since the model is using real train schedules, the time of a day is a determining factor not only for accurate train scheduling, but also for the amount of people using the line. Rush hour therefore presents much higher risk due to the sheer amount of people using the subway system in Tokyo. Cars with sarin arrived at the station around 8:00 A.M.

**5. Experiments**

The experiments aimed to show how the losses could be minimized in the future. There are, in fact, many more options for using the model. Simulations were performed in AnyLogic using the so-called "parameter variance" method. Thanks to this, it was possible to set parameters for the experiment and run the model several times without having to restart it. To minimize the randomness effect, the number of runs of each experiment was set to one hundred repeats, and the values were averaged.

Although the course of gas propagation is the same each time the model is launched, the differences in results are influenced, for example, by the random position or speed of individuals.

### 5.1. Experiment 1—Earlier Line Closure

The big problem in the attack was the late closure of the entire Hibiya line. It was closed, and all passengers were evacuated up to 35 min after the attack. Symptoms of sarin poisoning, however, occur after a few seconds of inhalation. If the evacuation occurred earlier, the consequences could be many times smaller.

For this experiment, the closure of the line was considered 10 min after the attack started. Since we only want to change the evacuation time, all other parameters of the initial model remained the same. Results are shown in Table 2.

**Table 2.** Results of Experiment 1.

|  | Light Exposure | Severe Exposure | Lethal Exposure |
|---|---|---|---|
| Average values | 115.49 | 3.26 | 1.61 |
| Maximal values | 150 | 8 | 5 |
| Minimal values | 86 | 0 | 0 |

As expected, the results showed a relatively significant difference from the baseline model. The number of people with light exposure is more than four times less in case of evacuation after 10 min. In the case of severe exposure, this is more than three times the difference. The lethal exposure brought exciting results. Considering the known 50% probability of death, an average of 1 person would die, after rounding. In an ideal scenario, the course could do without loss of life.

### 5.2. Experiment 2—More Frequent Train Departures

Human infection rate depends on two parameters. One is the distance from the source, and the other is the length of stay at a given location at a given density. It follows that one of the ways to minimize losses in a gas attack would be to reduce the length of time the passengers stayed at the attacked station. More frequent train arrivals could help. In the case of the Hibiya line, the average waiting time on the rail (Naka–Meguro) is around 3 min.

In the case of Kita–Senju, the average waiting time is about 2 min. It is one of the most frequented lines in the world. By default, trains run according to a real timetable. However, for the experiment, the waiting time for each rail was reduced by one more minute. In proportion to this change were also set parameters indicating the number of people leaving the station. The results of the second experiment are shown in Table 3.

**Table 3.** Results of Experiment 2.

|  | Light Exposure | Severe Exposure | Lethal Exposure |
|---|---|---|---|
| Average values | 334.97 | 5.18 | 0.71 |
| Maximal values | 380 | 13 | 4 |
| Minimal values | 293 | 1 | 0 |

Increasing the frequency of train arrivals by just one minute had a significant effect on the outcome of the infection. The number of people exposed to severe exposure decreased by more than half. One person was exposed to lethal exposure, which means there is a 50% chance that no one would die in this case.

### 5.3. Experiment 3—Partial Closure of the Platform

The density of sarin depends on the distance of the site from the source. Therefore, it would be desirable to prevent access to the immediate vicinity of the source in case of suspicion. Sarin packages were kicked out of the third car, i.e., almost at the end of one of the train platforms.

In this experiment, closure of part of the station from the entrance to the other side of the spreading source was considered. The time was determined as in the first experiment, so it is considered to close part of the platform 10 min from the start of the attack. Figure 5 shows the source of propagation and the considered closed part of the platform. The results of the third experiment are shown in Table 4.

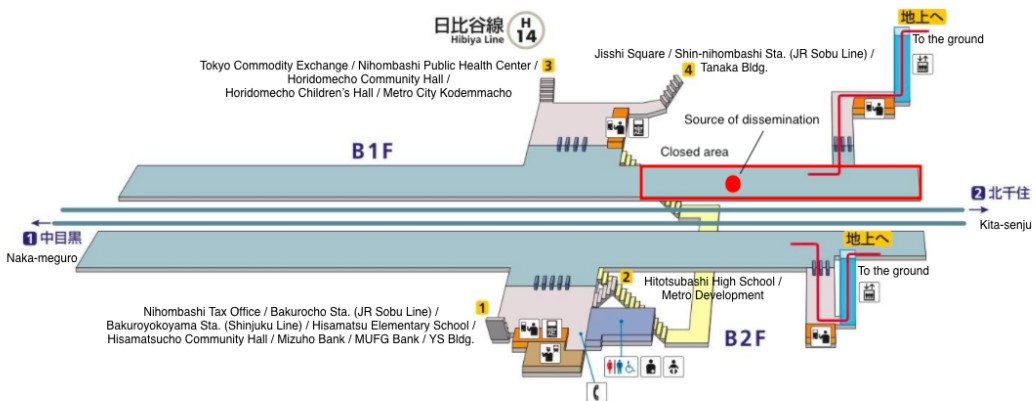

**Figure 5.** Experiment 3 (closing the platform).

**Table 4.** Results of Experiment 3.

|                | Light Exposure | Severe Exposure | Lethal Exposure |
| -------------- | -------------- | --------------- | --------------- |
| Average values | 157.74         | 3.18            | 1.77            |
| Maximal values | 201            | 7               | 6               |
| Minimal values | 127            | 0               | 0               |

The results of the experiment showed that if a dangerous gas leak is suspected, even a partial closure of the platform or any space is sufficient for a relatively fundamental difference in the number of infected people. Interestingly, given the position of the closed part and the gas source, it is clear that even delimiting the suspect packets in a 5-m perimeter would have a significant impact on the course of the attack and would greatly help to minimize losses.

### 5.4. Comparison

We performed a one-way ANOVA which rejected null hypothesis that all of population means are equal. Parameters and results of one-way ANOVA are as follows $\alpha = 0.05$, $k = 4$, $df = 96$, $P - value \approx 2.5 * 10^{-23}$. Additional Tukey–Kramer Post-Hoc tests showed that all three what-if scenarios significantly improves fatality results of the original model. Parameters for the test were $\alpha = 0.05$, $q - crit = 3.7$ and results for original model compared to all three experiments $q_1 = 15.1$, $q_2 = 17.2$ and, $q_3 = 16.4$ respectively.

### 5.5. Validation

No source provides sufficiently dissected data on injuries at the specific stations. According to [25] the Hibiya line had 2475 injuries and seven fatalities, from which four of them were at our station of interest. Four deaths at Kodemmacho station are therefore the only confirmed data we can use for validation of the basic simulation.

### 6. Discussion

For the model's accuracy of the simulation, it is essential to have enough attack information. Most of the crucial information was traceable, and the model achieved quite interesting results.

Nevertheless, some degree of abstraction was required compared to the real attack. In this discussion, some of the model shortcomings and future expansion options will be analyzed to achieve even greater accuracy.

One of the main things to be refined in the future is the spread of sarin through the environment. The values of sarin density at individual distances from the source correspond to the values estimated from the real attack, which allows relatively accurate monitoring of a particular contagion. Still, the spread itself is based on empirical data and assumptions. For the gas propagation to correspond to reality, it would be necessary to use relatively complex dispersion models.

Another issue that would make the model more precise is influencing the spread of gas through the environment, the ventilation, pedestrian or train movements. For example, to collect ventilation location information, etc. however, it would be necessary to visit the station in person. Such details are nowhere to be found and, of course, the metro management itself refuses to disclose them.

The model could also be refined by including some information about individual people moving around the station. For example, the health condition or age of an individual are properties that can affect the action of gas on the body.

## 7. Conclusions

A model was created to accurately simulate the course of the attack in Tokyo in 1995 at the metro station Kodemmacho. Three scenarios that could help minimize losses in a similar attack were considered, and experiments were performed. These experiments demonstrate the possibilities of using the model. At the end of the discussion, some shortcomings of the model and options of its extension in the future were mentioned.

**Author Contributions:** Conceptualization, O.D. and H.T.; methodology, H.T.; software, O.D.; validation, O.D. and H.T.; formal analysis, O.D.; investigation, O.D. and H.T.; resources, O.D. and H.T.; data curation, O.D.; writing—original draft preparation, O.D. and H.T.; writing—review and editing, O.D. and H.T.; visualization, O.D.; supervision, H.T. All authors have read and agreed to the published version of the manuscript.

**Funding:** This work was supported by a GACR 18-01246S and the Faculty of Informatics and Management UHK Specific Research Project.

**Acknowledgments:** This work was supported by a GACR 18-01246S and the Faculty of Informatics and Management UHK Specific Research Project. The authors would like to thank T. Nosek for his cooperation on this topic.

**Conflicts of Interest:** The authors declare no conflict of interest.

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
