# Peer review of "An Agent-Based Simulation to Minimize Losses during a Terrorist Attack"

_applsci, doi:10.3390/app10093213_

Round 1

Reviewer 1 Report

The authors simulate how the Sarin attack effects could have been mitigated by providing 3 improved scenarios.

The simulation seems accurate but many aspects could not be taken into account, some because of lack of data.

Calculating the infection based on Sarin density seems to be not very accurate, as people accumulate Sarin, for example in their lungs, so I would have preferred a function of concentration AND exposure time. Anyway the present form gives an idea of possible responses to the attack.

Another (minor) issue I would consider is that concentration drops after the parcels empty and the "cloud" spreads.

An interesting addition for further improvements (that I read is not possible with real values, at least at present) is the effect of ventilation as a way to remove the agent. A 2-3 cases with different ventilation values could be an interesting suggestion to give to subway builders, baying due attention on how Sarin is dispersed on the neighbor atmosphere.

Author Response

Thanks to the reviewer for his valuable comments and recommendations. The mentioned suggestions for improvement are for us a helpful inspiration for further extension. Thank you for your evaluation and for reviewing our article.

Reviewer 2 Report

  1. It is the responsibility of the authors to present their works in the best possible way. It is not the responsibility of the reader to find out that the paper as a valuable contribution despite a poor presentation. Please reconsider about that this manuscript has been well written to present your works. For example, Section 1.2. Basic models should be in the second section, Related Works.
  2. No ODD (Overview, Design concepts, Details) protocol was provided. An ODD is a community standard and must be included before this can be published. The modeling assumptions and model description should be explained as a similar manner to ODD to boost the reproducibility and replicability.

Grimm, V., Berger, U., Bastiansen, F., Eliassen, S., Ginot, V., Giske, J., ... & Huth, A. (2006). A standard protocol for describing individual-based and agent-based models. Ecological modelling, 198(1-2), 115-126.

  1. The ABMs need to be validated. Without validation, how can we trust the simulated outcomes? For this case, the authors can follow the Pattern-Oriented Modeling paradigm. At least you could compare the number of people at the station in your model to that in real-world.

Grimm, V., Revilla, E., Berger, U., Jeltsch, F., Mooij, W. M., Railsback, S. F., ... & DeAngelis, D. L. (2005). Pattern-oriented modeling of agent-based complex systems: lessons from ecology. science310(5750), 987-991.

Kang, J. Y., & Aldstadt, J. (2019). Using multiple scale spatio-temporal patterns for validating spatially explicit agent-based models. International Journal of Geographical Information Science33(1), 193-213.

Railsback, S. F., & Johnson, M. D. (2011). Pattern-oriented modeling of bird foraging and pest control in coffee farms. Ecological Modelling222(18), 3305-3319.

  1. How many simulation runs did you performed? Due to the stochastic nature of ABM, a small number of simulation runs does not help to capture the patterns of simulated outcomes.

  1. In general, to compare the scenario, you could try to statistical tests, such as ANOVA. Then you can easily present which experiments have better effects on minimizing the losses.

Author Response

Thanks to the reviewer for his valuable comments and recommendations. The mentioned suggestions for improvement are for us a helpful inspiration. Thank you for your evaluation and for reviewing our article.

Comment 1

It is the responsibility of the authors to present their works in the best possible way. It is not the responsibility of the reader to find out that the paper as a valuable contribution despite a poor presentation. Please reconsider about that this manuscript has been well written to present your works. For example, Section 1.2. Basic models should be in the second section, Related Works.

Answer 1

Thanks to the reviewer for his valuable comments and recommendations. This part of the text has been moved to the Related works section. The presentation of the content has been improved in this revised version of the text.

Comment 2:

No ODD (Overview, Design concepts, Details) protocol was provided. An ODD is a community standard and must be included before this can be published. The modeling assumptions and model description should be explained as a similar manner to ODD to boost the reproducibility and replicability.

Grimm, V., Berger, U., Bastiansen, F., Eliassen, S., Ginot, V., Giske, J., ... & Huth, A. (2006). A standard protocol for describing individual-based and agent-based models. Ecological modelling, 198(1-2), 115-126.

Answer 2

Thanks to the reviewer for his valuable comments and recommendations. The mentioned part of the text was rewritten and expanded according to the recommended structure of the cited publication.

Comment 3

The ABMs need to be validated. Without validation, how can we trust the simulated outcomes? For this case, the authors can follow the Pattern-Oriented Modeling paradigm. At least you could compare the number of people at the station in your model to that in real-world.

  • Grimm, V., Revilla, E., Berger, U., Jeltsch, F., Mooij, W. M., Railsback, S. F., ... & DeAngelis, D. L. (2005). Pattern-oriented modeling of agent-based complex systems: lessons from ecology. science, 310(5750), 987-991.
  • Kang, J. Y., & Aldstadt, J. (2019). Using multiple scale spatio-temporal patterns for validating spatially explicit agent-based models. International Journal of Geographical Information Science, 33(1), 193-213.
  • Railsback, S. F., & Johnson, M. D. (2011). Pattern-oriented modeling of bird foraging and pest control in coffee farms. Ecological Modelling, 222(18), 3305-3319.

Answer 3

Thanks to the reviewer for his valuable comments and recommendations. We have added a section dealing with model validation.

Comment 4

How many simulation runs did you performed? Due to the stochastic nature of ABM, a small number of simulation runs does not help to capture the patterns of simulated outcomes.

Answer 4

Thanks to the reviewer for his valuable comments and recommendations. To minimize the effect of randomness, the number of runs of each experiment was set to one hundred replicates, and the values were arithmetically averaged. This note is listed in the Experiments section.

Comment 5

In general, to compare the scenario, you could try to statistical tests, such as ANOVA. Then you can easily present which experiments have better effects on minimizing the losses. 

Answer 5

Thanks to the reviewer for his valuable comments and recommendations. The Comparison section has been added to the text, where we comment on the statistical assessment and comparison of experiments.

Round 2

Reviewer 2 Report

Thank you for your revision. 

You should present the results from ANOVA and Tukey Post-Hoc tests (Section 5.4). Otherwise, I am not for sure about how you performed those.

Author Response

Comment 1

You should present the results from ANOVA and Tukey Post-Hoc tests (Section 5.4). Otherwise, I am not for sure about how you performed those.

Answer 1

Thanks to the reviewer for his valuable comments. We performed One-way ANOVA which rejected the null hypothesis that all of the population means are equal. Parameters and results of one-way ANOVA are as following  alpha = 0.05, k = 4, df = 96, P-value ≈ 2.5e-23. Additional Tukey-Kramer Post-Hoc test showed that all three what-if scenarios significantly improves fatalities results of the original model. Parameters for the test were alpha = 0.05, q-crit = 3.7 and results for original model compared to all three experiments q1 = 15.1, q2 = 17.2 and, q3 = 16.4 respectively.